# Metal-Free Catalyzed Oxidation/Decarboxylative [3+2] Cycloaddition Sequences of 3-Formylchromones to Access Pyrroles with Anti-Cancer Activity

**DOI:** 10.3390/molecules28227602

**Published:** 2023-11-15

**Authors:** Xue Li, Xing-Yu Chen, Bing-Ying Fan, Qun Yu, Jie Lei, Zhi-Gang Xu, Zhong-Zhu Chen

**Affiliations:** 1College of Pharmacy, National & Local Joint Engineering Research Center of Targeted and Innovative Therapeutics, IATTI, Chongqing University of Arts and Sciences, Chongqing 402160, China; lxshirley@cqwu.edu.cn (X.L.); 15320378927@163.com (X.-Y.C.); 15310400564@163.com (B.-Y.F.); 15084401382@163.com (Q.Y.); 2Chongqing Key Laboratory of Natural Product Synthesis and Drug Research, School of Pharmaceutical Sciences, Chongqing University, Chongqing 400044, China; 3Institute of Bioorganic & Medicinal Chemistry, School of Chemistry and Chemical Engineering, Southwest University, Chongqing 400715, China; 4Chongqing Academy of Chinese Materia Medica, Chongqing 400067, China

**Keywords:** metal-free, decarboxylation, 3-formylchromones, pyrroles, anti-cancer activity

## Abstract

An efficient and direct approach to pyrroles was successfully developed by employing 3-formylchromones as decarboxylative coupling partners, and facilitated by microwave irradiation. The protocol utilizes easily accessible feedstocks, a catalytic amount of DBU without any metals, resulting in high efficiency and regioselectivity. Notably, all synthesized products were evaluated against five different cancer cell lines and compound **3l** selectively inhibited the proliferation of HCT116 cells with an IC_50_ value of 10.65 μM.

## 1. Introduction

Substituted pyrroles are a valuable class of five-membered nitrogen heterocycles and are considered as “privileged scaffolds” found in natural products and bioactive compounds (Figure 1) [1,2,3,4]. Molecules containing the pyrrole unit have been demonstrated to have a broad range of biological properties, including being a tubulin polymerization inhibitor [5], a Cdc7 kinase inhibitor [6], a RTK inhibitor (Sunitinib) [7,8,9], a potassium-competitive acid blocker (P-CAB) [10], a natural antibiotic (pyoluteorin) [11,12], a dual PPARα/γ agonist (saroglitazar) [13,14], and marineosin [15].

Decarboxylative cyclization reactions are highly efficient strategies for synthesizing high-value organic molecules through the formation of C-C bond to form heterocycles, which serve as crucial structural motifs in a wide range of natural products and biologically active compounds [16,17]. Recently, a palladium-catalyzed three-component decarboxylation of propargylic carbonate used to produce pyrroles was described by Zhu and co-workers [18]. Cao et al. reported decarboxylative cycloaddition of β-ketoacids for synthesis of pyrrole by employing iron as catalysts through a Mannich/decarboxylative process [19]. However, the majority of representative decarboxylative cascades for synthesizing heterocyclic molecules have encountered at least one drawback: (1) a metal catalysis was required; (2) a large excess of additive was necessary; or (3) a highly activated substrate was involved (Figure 1a). Therefore, a method capable of operating with a broad functional group tolerance and employing a catalytic amount of additive without transitional metals would be highly desirable.

Pyrrole plays a crucial role in heterocyclic compounds due to its biological significance. The synthesis of substituted pyrroles is a significant area of research for rapidly expanding the diversity of pyrroles. Recently, Yang and colleagues reported the Ag_2_CO_3_-catalyzed cycloaddition for the synthesis of pyrroles under mild reaction conditions [20]. Furthermore, Yang’s group observed the silver-promoted cascade cyclization of 3-I-substituted chromones and ethyl isocyanoacetate, leading to the formation of chromeno[2,3-*b*]pyrrol-4(1*H*)-ones in good yields [21]. Zhao and co-workers presented Ag-catalyzed unusual transformations that enabled the synthesis of polysubstituted pyrroles through three-component reactions involving isocyanoacetates, amines, and 3-formylchromones [22]. In terms of the powerful and versatile synthon of 3-formylchromone, great efforts for synthesis of pyrroles have been made by Sliva and Ryabukhin’s group [23,24]. Recently, Kamal explored the combination of chromones and phenacyl azides using an aza-Wittig reaction [25]. The mixture of 3-formylchromones and *N*-phenyl glycines delivered functionalized chromeno[2,3-*b*]pyrrol-4(1*H*)-ones with CuBr as the catalyst [26]. To date, no methodology involving decarboxylation of 3-formylchromones for the synthesis of substituted pyrroles has been reported.

Over the past few years, we have been focusing on developing diverse heterocycle synthesis using a 3-formylchromone core [27,28,29]. Chromone, an undoubtedly activated Michael acceptor and a high-value starting material, has been the subject of our interest. We hypothesized that 3-formylchromone could serve as a precursor to chromone-3-carboxylic acid and directly yield substituted pyrroles. In this work, we will report on the DBU-driven [3+2] cycloaddition sequence promoting the synthesis of substituted pyrrole through oxidation, Michael reaction, and decarboxylation steps. These reactions were performed under microwave reaction conditions (Figure 1b). The significance of this chemistry can be summarized as follows: (1) no metal catalysts; (2) no stoichiometric additives; (3) high regioselectivities; and (4) the generation of pyrroles can be achieved in just 10 min, affording moderate to good yield of products.

## 2. Results

To test our hypothesis, we selected the reaction between 3-formylchromone **1a** and toluenesulphonylmethyl isocyanide (TosMIC) **2a** as the model substrates. The reaction was carried out with DCE (1,2-dichloroethane) as the solvent. Microwave technology is a highly efficient heating technique for the preparation of bioactive compounds in organic chemistry, owing to its short reaction times, higher yields, and good regioselectivity [30]. Moreover, taking inspiration from the fact that microwave-promoted decarboxylative reactions have been successfully applied to various substrates under optimized conditions [31,32], we performed the reaction under microwave assistance (Table 1, entry 1). The results showed that 1.0 equiv. of K_2_CO_3_ produced the desired product **3a** with a yield of 27%. However, most of the starting materials were left, which forced us to try other bases. After a brief screening of other basic conditions (entries 1–7), we were pleased to find that DBU (1,8-diazabicyclo[5.4.0]undec-7-ene) resulted in the formation of **3a** with a yield of 45%. These results suggested that a transition metal was not required for the decarboxylation of 3-formylchromones to produce substituted pyrroles.

Encouraged by these promising results, we examined the solvents DMSO (dimethyl sulfoxide), DMF (*N,N*-dimethylformamide), toluene, THF (tetrahydrofuran), MeCN (acetonitrile), EtOH (ethanol), and 1,4-dioxane to enhance the conversion of starting materials (entries 8–15). Following the GSK solvent sustainability guide [33], 1-hexanol and *N*-ethylpyrrolidone were additionally employed in our method. No satisfactory yield of **3a** was obtained (entries 16-17). Among the screened the solvents, MeCN obtained the substituted pyrroles with the higher yields (entry 13). When the model reaction was treated with reaction conditions using a catalytic amount of DBU, the higher yield of **3a** was obtained (entry 20). This result indicated the decarboxylation reaction did not require a transition metal and the large excess of base was not essential to directly generate substituted pyrroles. Under otherwise identical conditions, by elevating reaction temperature from 80 °C to 120 °C, it was found that base-triggered oxidation/decarboxylation of 3-formylchromones worked efficiently under microwave irradiation at 100 °C (entries 18–20). Additionally, prolonged and shortened reaction times did not afford satisfactory yields (entries 21 and 22). To further confirm the powerful microwave irradiation technology, the model reaction was conducted with conventional heating at 100 °C for 1 h, resulting in a 51% yield of the desired product **3a**. Therefore, the optimized reaction conditions were determined to be: 3-formylchromone (0.3 mmol), isocyanide (1.0 equiv.), DBU (30 mol%), MeCN (3.0 mL), microwave irradiation at 100 °C for 10 min.

Using the optimized reaction conditions, we evaluated the generalizability of DBU-mediated direct oxidation/decarboxylation for accessing substituted pyrroles by utilizing various 3-formylchromones and isocyanides, as shown in Figure 2. In contrast, DBU-driven oxidation/decarboxylative [3+2] cycloaddition exhibited high chemoselectivity, and no 2-substituted chromanones were observed [34]. Under the standard reaction conditions, the yields of the 2-tosylxy-4-benzolyoxy pyrroles **3a**–**3p** ranged from 60% to 78%. When TosMIC was used as one of the starting materials, the 3-formylchromones containing electron-donating groups (H, Me, Et, *i*-Pr, OCOMe) were efficiently converted into the desired compounds **3a**–**3e**. Furthermore, the 3-formylchromones containing electron-withdrawing groups (F, Cl, di-Cl, Br) were also tolerated and yielded the final products in moderate to good yields under the optimized reaction conditions. Specifically, compound **3g** (CCDC 2282072, see Figure 2) was isolated and its structure was determined by X-ray analysis, as shown in Figure 2. However, using 6-nitrochromone-3-carboxaldehyde for the synthesis of compound **3k** afforded a diminished yield. To our surprise, the reported protocols for pyrrole synthesis commonly utilize either methyl isocyanoacetate or TosMIC. To the best of our knowledge, isocyanoacetate exhibits similar properties to TosMIC in producing pyrrole analogues. In our protocol, isocyanoacetate served as an effective synthon for generating the 2-methoxycarbonyl-3-benzoyloxy pyrroles through decarboxylative [3+2] cycloaddition (**3l**–**3p**). The different substitutions on the phenyl ring of chromone-3-carboxaldehyde have little or no effect on the yield. Regrettably, DBU-mediated oxidation/decarboxylative [3+2] cycloaddition using benzyl isocyanide with inactive methylene as a substrate for obtaining compound **3q** was found to be unfeasible, as shown in Figure 2.

In order to validate the mechanistic details, a series of control experiments was conducted. Firstly, the model substrates **1a** and **2a** were reacted in the presence of 3.0 equiv. of TEMPO, resulting in the isolation of the cycloaddition adduct **3a** with a yield of 68%. This observation suggests that the radical pathway for the synthesis of **3a** could be excluded (Figure 3a). To elucidate the mechanism of the aldehyde group removed from 3-formylchromone, a control experiment was performed using chromone-3-carboxylic acid in Figure 3b. Compound **4** led to the formation of the desired product **3a** with a yield of 64%. Therefore, we concluded that the oxidation of the aldehyde could be followed by decarboxylative [3+2] cycloaddition. As shown in Figure 3c, aerobic oxidation is a crucial step in obtaining the final products. The yield of final product **3a** was increased to 76% under atmosphere, which proved that the oxidant for the conversion of **1a** to **4** is oxygen in air. Subsequently, in order to elucidate the decarboxylation pathway, the reaction between chromone **5** and TosMIC **2a** was conducted under standard reaction conditions, resulting in the detection of a trace amount of **3a** (Figure 3d). Therefore, chromone is not a key intermediate in this process.

Based on the experimental results presented above and relevant reports in the literature, a possible mechanistic pathway for the formation of pyrroles is proposed in Figure 4. 3-Formylchromone is oxidized using microwave irradiation with air as the exclusive oxidant. Under basic reaction conditions, deprotonation of **2a** gives rise to the nucleophilic carbanion **6**. Intermediate **7** is formed through a *retro*-Michael addition reaction, followed by γ-pyrone ring opening to yield intermediate **8** [35]. This structure **8** further undergoes deprotonation for intermediate **9** and decarboxylation to yield intermediate **10**. To the best of our knowledge, isocyanide is a well-known zwitterionic compound that can act as a nucleophile trap. The isocyano group of **10** undergoes intramolecular α-nucleophilic addition to yield the pyrrole precursor **11** [36]. Finally, the 1,2-*H* shift process of compound **11** leads to the aromatization and the formation of the desired product **3a**.

Pyrroles are significant scaffolds in various bioactive compounds and have diverse biological applications. To assess the potential of developing a drug candidate from all the synthesized pyrroles, we selected a panel of human tumor cell lines (A549, DU145, H8, PC3, and HCT116), which are classified as difficult-to-inhibit cell lines in the National Cancer Institute’s panel of 60 human cancer cell lines (Figure 3). To our delight, **3l** exhibited potent antitumor activities in vitro, with an IC_50_ value of 10.65 μM against HCT116 cells. Also, colony formation assay confirmed the results that smaller and lesser colonies were formed in PC3 cells treated with compound **3l**, compared with a control group after exposure to inhibitors for 10 days. These assays suggest that compound **3l** inhibited the proliferation and growth of HCT116 cell lines in vitro at micromolar concentrations. Moreover, **3l** could be regarded as a lead compound for further modification in the discovery of an anticancer agent.

## 3. Experimental Section

### 3.1. General Information

Each component of laboratory glassware was oven-dried, then used for carrying out the general experimental procedures. ^1^H and ^13^C NMR were recorded on a Bruker 400 spectrometer, frequency: 6–440 MHz; frequency resolution: ≤0.005 Hz; sensitivity: ^1^H ≥ 500:1, ^13^C ≥ 220:1, ^19^F ≥ 500:1, probe: 5 mm BBFO Z-gradient. ^1^H NMR data are reported as follows: chemical shift in ppm (δ), multiplicity (s = singlet, d = doublet, t= triplet, m = multiplet), coupling constant (Hz), relative intensity. ^13^C NMR data are reported as follows: chemical shift in ppm (δ). HPLC-MS analyses were performed on a Shimadzu-2020 LC-MS instrument using the following conditions: Shim-pack VP-ODS C18 column (reverse phase, 150 × 2.0 mm); 90% acetonitrile and 10% water over 6.0 min; flow rate of 0.5 mL/min; UV photodiode array detection from 200 to 400 nm. High-resolution mass spectra (HRMS) were recorded on a Q Exactive hybrid quadrupole-Orbitrap mass spectrometer (Bremen, Germany, Thermo Fisher Scientific) with an ESI source of 140,000 fwhm, the AGC target set to 1 × 106, and a scan range of 100–1000 *m*/*z*. The raw data were deconvoluted using Xcalibur 4.1. The products were purified by Biotage Isolera™ Spektra Systems and Hexane/EtOAc solvent systems. All reagents and solvents were obtained from commercial sources and used without further purification. 

### 3.2. Microwave Irradiation Experiments 

All microwave irradiation experiments were carried out in a Biotage Initiator Classic microwave apparatus with continuous irradiation power from 0 to 400 W with utilization of the standard absorbance level of 250 W maximum power. The reactions were carried out in 10 mL glass tubes, sealed with a microwave cavity. The reaction was irradiated at the required ceiling temperature (the reaction temperature was monitored by an external surface sensor using the Biotage Initiator reactor) using maximum power for the stipulated time. Then, it was cooled to 50 °C with gas jet cooling.

### 3.3. Cell Lines and Culture and Viability Assay

Cell lines and culture. The human lung cancer cell A549, prostate cancer cell DU145, cervical epithelial cell H8, prostate cancer cell PC3, and colon carcinoma cells HCT116 were purchased from American Type Culture Collection (ATCC, Manassas, VA, USA). The cancer cells were cultured in high-glucose DMEM (Hyclone, SH30022.01, Logan, UT, USA) medium supplemented with 10% fetal bovine serum (FBS, Gibco, 10099, Thomastown, Australia origin). The cells were maintained in the incubator at 37 °C and 5% CO_2_ with humidified atmosphere.

MTT assay. Cancer cells were counted and seeded into the 96-well plate containing 100 µL complete medium; the density of the cells was 3 × 10^3^ cells per well. After incubation for 24 h, another 100 µL complete medium containing 10 µM compounds **3** or equal amount of dimethyl sulfoxide (DMSO) was added; each treatment was triple replicated. The compound-treated cells were cultured for another 48 h, 3-(4, 5-dimethyl-2-thiazolyl)-2,5-diphenyl-2-*H*-tetrazolium bromide (MTT, Beyotime, ST316, Shanghai, China) was added, and the plate was incubated for another 4 h. After incubation, the medium was removed and 150 μL DMSO was added into each well to dissolve the formazan. The optical density (OD) of each well was measured with a microplate reader (Bio-Tek, Winooski, VT, USA) at an absorbance wavelength of 570 nm. The viability of the compound-treated cells equals the ratio of OD_compound_ to OD_DMSO_.

Colony formation assay. HCT116 cells were harvested at a density of 80% and seeded onto the 6-well plates with 400 cells per well containing 1 mL medium. After 24 h incubation, HCT116 cells were treated with compound **3l** with different concentrations (0, 5, 10, and 20 μM) for 48 h, then changed to a fresh medium without compound **3l** and incubated for another 10 days. After that, HCT116 cells were washed with PBS three times and fixed with 4% paraformaldehyde for 25 min. Finally, the cells were further washed with PBS three times again, and stained with a 0.5% crystal violet staining solution (Beyotime, C0121, Shanghai, China).

### 3.4. Synthetic Procedures for the Synthesis of Compound 3

3-Formylchromone (52 mg, 0.3 mmol), isocyanide (0.3 mmol), and DBU (14 mg, 30 mol %) were dissolved in MeCN (3.0 mL) in a 5 mL microwave vial. Then, the mixture was sealed and heated under microwave irradiation at 100 °C for 10 min. The reaction mixture was then cooled to room temperature and concentrated under reduced pressure to give a residue, which was diluted with EtOAc (15.0 mL), before being washed sequentially with saturated brine. The organic solution was then dried over MgSO_4_ and concentrated to give a residue, which was purified by column chromatography over silica gel eluting with a gradient of ethyl acetate/hexane (0−50%) to afford the relative pyrrole compound **3**.

*(2-Hydroxyphenyl)(5-tosyl-1H-pyrrol-3-yl)methanone* (**3a**), a white solid (72%), R_f_ = 0.25 (n-Hexane/EtOAc, 8:2), m.p. = 135–136 °C. ^1^H NMR (400 MHz, CDCl_3_) δ 11.80 (s, 1H), 10.22 (s, 1H), 7.78 (d, J = 8.3 Hz, 2H), 7.74 (dd, J = 8.0, 1.5 Hz, 1H), 7.48 (dd, J = 3.1, 1.6 Hz, 1H), 7.45 − 7.38 (m, 1H), 7.28 − 7.20 (m, 3H), 6.98 − 6.93 (m, 1H), 6.88 − 6.81 (m, 1H), 2.34 (s, 3H). ^13^C NMR (100 MHz, CDCl_3_) δ 193.14, 162.55, 144.90, 138.24, 135.96, 131.65, 130.85, 130.18, 127.92, 127.19, 125.41, 119.76, 119.00, 118.43, 116.28, 21.64. HRMS (ESI) m/z: [M+H]^+^ Calcd for C_18_H_16_NO_4_S^+^ 342.0795; Found 342.0795.

*(2-Hydroxy-5-isopropylphenyl)(5-tosyl-1H-pyrrol-3-yl)methanone* (**3b**), a white solid (78%), R_f_ = 0.25 (n-Hexane/EtOAc, 8:2), m.p. = 138–139 °C. ^1^H NMR (400 MHz, CDCl_3_) δ 11.56 (s, 1H), 9.78 (s, 1H), 7.78 (d, J = 8.3 Hz, 2H), 7.53 (d, J = 2.2 Hz, 1H), 7.45 (dd, J = 3.2, 1.6 Hz, 1H), 7.31 (dd, J = 8.6, 2.3 Hz, 1H), 7.26 (d, J = 8.1 Hz, 2H), 7.17 (dd, J = 2.5, 1.7 Hz, 1H), 6.90 (d, J = 8.6 Hz, 1H), 2.80 (dt, J = 13.8, 6.9 Hz, 1H), 2.36 (s, 3H), 1.15 (d, J = 6.9 Hz, 6H). ^13^C NMR (100 MHz, CDCl_3_) δ 193.16, 160.64, 144.85, 139.26, 138.33, 134.52, 131.02, 130.15, 128.84, 127.25, 125.70, 119.46, 118.23, 116.11, 33.22, 24.05, 21.64. HRMS (ESI) m/z: [M+H]^+^ Calcd for C_21_H_22_NO_4_S^+^ 384.1265; Found 384.1262.

*(2-Hydroxy-5-methylphenyl)(5-tosyl-1H-pyrrol-3-yl)methanone* (**3c**), a white solid (75%), R_f_ = 0.23 (n-Hexane/EtOAc, 8:2), m.p. = 134–135 °C. ^1^H NMR (400 MHz, DMSO-d6) δ 13.10 (s, 1H), 10.54 (s, 1H), 7.86 (d, J = 8.1 Hz, 2H), 7.63 (s, 1H), 7.44 (d, J = 8.1 Hz, 2H), 7.31 (s, 1H), 7.24 (d, J = 8.2 Hz, 1H), 7.13 (s, 1H), 6.86 (d, J = 8.3 Hz, 1H), 2.38 (s, 3H), 2.24 (s, 3H). ^13^C NMR (100 MHz, DMSO-d6) δ 191.36, 156.01, 144.75, 139.03, 134.62, 131.03, 130.86, 130.57, 128.21, 127.42, 125.25, 124.25, 117.35, 116.06, 21.50, 20.42. HRMS (ESI) m/z: [M+H]^+^ Calcd for C_19_H_18_NO_4_S^+^ 356.0952; Found 356.0953.

*(5-Ethyl-2-hydroxyphenyl)(5-tosyl-1H-pyrrol-3-yl)methanone* (**3d**), a white solid (77%), R_f_ = 0.25 (n-Hexane/EtOAc, 8:2), m.p. = 137–138 °C. ^1^H NMR (400 MHz, CDCl_3_) δ 11.59 (s, 1H), 9.86 (s, 1H), 7.78 (d, J = 8.3 Hz, 2H), 7.52 (d, J = 2.1 Hz, 1H), 7.46 (dd, J = 3.1, 1.6 Hz, 1H), 7.30 − 7.20 (m, 4H), 6.89 (d, J = 8.5 Hz, 1H), 2.53 (q, J = 7.6 Hz, 2H), 2.35 (s, 3H), 1.14 (t, J = 7.6 Hz, 3H). ^13^C NMR (100 MHz, CDCl_3_) δ 193.14, 160.61, 144.86, 138.30, 135.89, 134.62, 130.20, 127.52, 127.23, 125.63, 119.52, 118.26, 116.17, 27.96, 21.65, 15.73. HRMS (ESI) m/z: [M+H]^+^ Calcd for C_20_H_20_NO_4_S^+^ 370.1108; Found 370.1109.

*4-Hydroxy-3-(5-tosyl-1H-pyrrole-3-carbonyl)phenyl acetate* (**3e**), a white solid (75%), R_f_ = 0.21 (n-Hexane/EtOAc, 8:2), m.p. = 133–134 °C. ^1^H NMR (400 MHz, CDCl_3_) δ 12.05 (s, 1H), 9.92 (s, 1H), 7.77 (d, J = 8.2 Hz, 3H), 7.46 (s, 1H), 7.26 (d, J = 8.1 Hz, 2H), 6.71 (d, J = 2.1 Hz, 1H), 6.62 (dd, J = 8.7, 2.1 Hz, 1H), 2.35 (s, 3H), 2.26 (s, 3H). ^13^C NMR (100 MHz, CDCl_3_) δ 192.26, 168.68, 164.19, 156.23, 144.92, 138.24, 132.88, 131.11, 130.19, 127.69, 127.24, 125.31, 116.04, 112.91, 111.27, 21.65, 21.22. HRMS (ESI) m/z: [M+H]^+^ Calcd for C_20_H_18_NO_6_S^+^ 400.0850; Found 400.0851.

*(4-Chloro-2-hydroxy-5-methylphenyl)(5-tosyl-1H-pyrrol-3-yl)methanone* (**3f**), a white solid (72%), R_f_ = 0.23 (n-Hexane/EtOAc, 8:2), m.p. = 131–132 °C. ^1^H NMR (400 MHz, CDCl_3_) δ 11.69 (s, 1H), 10.17 (s, 1H), 7.79 (d, J = 8.3 Hz, 2H), 7.66 (s, 1H), 7.46 (dd, J = 3.0, 1.6 Hz, 1H), 7.26 (d, J = 8.2 Hz, 2H), 7.20 (d, J = 2.2 Hz, 1H), 6.84 (s, 1H), 2.34 (s, 3H), 2.31 (s, 3H). ^13^C NMR (100 MHz, CDCl_3_) δ 191.65, 161.02, 145.13, 144.95, 138.20, 131.20, 130.95, 130.19, 127.68, 127.22, 125.06, 124.30, 120.49, 118.67, 116.00, 21.61, 20.71. HRMS (ESI) m/z: [M+H]^+^ Calcd for C_19_H_17_ClNO_4_S^+^ 390.0562; Found 390.0560.

*(5-Chloro-2-hydroxyphenyl)(5-tosyl-1H-pyrrol-3-yl)methanone* (**3g**), a white solid (74%), R_f_ = 0.23 (n-Hexane/EtOAc, 8:2), m.p. = 133–134 °C. ^1^H NMR (400 MHz, CDCl_3_) δ 11.66 (s, 1H), 10.28 (s, 1H), 7.79 (d, J = 8.3 Hz, 2H), 7.68 (d, J = 2.6 Hz, 1H), 7.47 (dd, J = 3.2, 1.6 Hz, 1H), 7.35 (dd, J = 8.9, 2.6 Hz, 1H), 7.27 (d, J = 8.2 Hz, 2H), 7.22 − 7.19 (m, 1H), 6.92 (d, J = 8.9 Hz, 1H), 2.35 (s, 3H). ^13^C NMR (100 MHz, CDCl_3_) δ 192.00, 160.98, 145.01, 138.12, 135.73, 131.40, 130.55, 130.22, 127.96, 127.24, 124.86, 123.69, 120.43, 120.08, 116.04, 21.65. HRMS (ESI) m/z: [M+H]^+^ Calcd for C_18_H_15_ClNO_4_S^+^ 376.0405; Found 376.0404.

*(5-Bromo-2-hydroxyphenyl)(5-tosyl-1H-pyrrol-3-yl)methanone* (**3h**), A white solid (76%), R_f_ = 0.25 (n-Hexane/EtOAc, 8:2), m.p. = 130–131 °C. ^1^H NMR (400 MHz, CDCl_3_) δ 11.67 (s, 1H), 10.31 (s, 1H), 7.79 (d, J = 7.8 Hz, 3H), 7.50 − 7.44 (m, 2H), 7.26 (d, J = 8.1 Hz, 2H), 7.22 − 7.19 (m, 1H), 6.86 (d, J = 8.9 Hz, 1H), 2.35 (s, 3H). ^13^C NMR (100 MHz, CDCl_3_) δ 191.90, 161.42, 145.04, 138.50, 138.08, 133.54, 131.37, 130.24, 128.02, 127.25, 124.84, 121.07, 120.48, 116.08, 110.56, 21.66. HRMS (ESI) m/z: [M+H]^+^ Calcd for C_18_H_15_BrNO_4_S^+^ 419.9900; Found 419.9900.

*(5-Fluoro-2-hydroxyphenyl)(5-tosyl-1H-pyrrol-3-yl)methanone* (**3i**), a white solid (71%), R_f_ = 0.23 (n-Hexane/EtOAc, 8:2), m.p. = 138–139 °C. ^1^H NMR (400 MHz, CDCl_3_) δ 11.49 (s, 1H), 9.85 (s, 1H), 7.78 (d, J = 8.3 Hz, 2H), 7.51 − 7.45 (m, 1H), 7.41 (dd, J = 8.8, 3.1 Hz, 1H), 7.27 (d, J = 8.2 Hz, 2H), 7.19 − 7.13 (m, 2H), 6.93 (dd, J = 9.1, 4.6 Hz, 1H), 2.36 (s, 3H). ^13^C NMR (100 MHz, CDCl_3_) δ 192.11, 158.70, 156.17, 153.68, 144.98, 138.17, 131.45, 130.20, 129.93, 127.55, 125.13, 123.50, 119.80, 116.62, 115.84, 21.65. HRMS (ESI) m/z: [M+H]^+^ Calcd for C_18_H_15_NO_4_S^+^ 360.0701; Found 360.0700.

*(4,5-Dichloro-2-hydroxyphenyl)(5-tosyl-1H-pyrrol-3-yl)methanone* (**3j**), a white solid (69%), R_f_ = 0.24 (n-Hexane/EtOAc, 8:2), m.p. = 140–141 °C. ^1^H NMR (400 MHz, DMSO-d6) δ 13.20 (s, 1H), 10.72 (s, 1H), 7.87 (d, J = 8.3 Hz, 2H), 7.73 (d, J = 2.6 Hz, 1H), 7.67 (dd, J = 3.3, 1.7 Hz, 1H), 7.50 − 7.38 (m, 3H), 7.20 − 7.09 (m, 1H), 2.38 (s, 3H). ^13^C NMR (100 MHz, DMSO-d6) δ 188.60, 151.69, 144.83, 138.91, 132.36, 131.70, 131.67, 130.57, 128.68, 128.31, 127.46, 124.82, 123.64, 123.34, 115.77, 21.51. HRMS (ESI) m/z: [M+H]^+^ Calcd for C_18_H_14_Cl_2_NO_4_S^+^ 410.0016; Found 410.0017.

*(2-Hydroxy-5-nitrophenyl)(5-tosyl-1H-pyrrol-3-yl)methanone* (**3k**), a white solid (60%), R_f_ = 0.17 (n-Hexane/EtOAc, 8:2), m.p. = 145–146 °C. ^1^H NMR (400 MHz, DMSO-d6) δ 13.14 (s, 1H), 11.57 (s, 1H), 8.25 (dd, J = 9.1, 2.8 Hz, 1H), 8.15 (d, J = 2.8 Hz, 1H), 7.86 (d, J = 8.2 Hz, 2H), 7.65 (s, 1H), 7.44 (d, J = 8.1 Hz, 2H), 7.12 (d, J = 9.0 Hz, 2H), 2.38 (s, 3H). ^13^C NMR (100 MHz, DMSO-d6) δ 187.87, 161.93, 144.80, 139.78, 138.95, 131.54, 131.48, 130.57, 128.02, 127.41, 125.82, 125.52, 117.71, 115.61, 21.50. HRMS (ESI) m/z: [M+H]^+^ Calcd for C_18_H_15_N_2_O_6_S^+^ 387.0646; Found 387.0645.

*Methyl 4-(2-hydroxybenzoyl)-1H-pyrrole-2-carboxylate* (**3l**), a white solid (61%), R_f_ = 0.30 (n-Hexane/EtOAc, 8:2), m.p. = 124–125 °C. ^1^H NMR (400 MHz, DMSO-d6) δ 12.72 (s, 1H), 10.94 (s, 1H), 7.60 (dd, J = 7.7, 1.6 Hz, 1H), 7.56 (dd, J = 3.4, 1.6 Hz, 1H), 7.44 (ddd, J = 8.7, 7.4, 1.7 Hz, 1H), 7.13 (dd, J = 2.4, 1.7 Hz, 1H), 7.01 − 6.88 (m, 2H), 3.81 (s, 3H). ^13^C NMR (100 MHz, DMSO-d6) δ 191.83, 160.97, 158.55, 134.01, 130.81, 129.91, 125.31, 124.33, 123.90, 119.51, 117.52, 116.36, 52.07. HRMS (ESI) m/z: [M+H]^+^ Calcd for C_13_H_12_NO_4_^+^ 246.0761; Found 246.0757.

*Methyl 4-(2-hydroxy-5-isopropylbenzoyl)-1H-pyrrole-2-carboxylate* (**3m**), a white solid (64%), R_f_ = 0.32 (n-Hexane/EtOAc, 8:2), m.p. = 128–129 °C. ^1^H NMR (400 MHz, CDCl_3_) δ 11.83 (s, 1H), 9.63 (s, 1H), 7.73 (d, J = 2.3 Hz, 1H), 7.59 (dd, J = 3.2, 1.6 Hz, 1H), 7.39 (ddd, J = 5.4, 4.0, 2.0 Hz, 2H), 7.01 (d, J = 8.5 Hz, 1H), 3.94 (s, 3H), 2.92 (dt, J = 13.8, 6.9 Hz, 1H), 1.27 (d, J = 6.9 Hz, 6H). ^13^C NMR (100 MHz, CDCl_3_) δ 193.69, 161.18, 160.64, 139.13, 134.15, 129.04, 127.46, 125.35, 123.92, 119.72, 118.15, 116.45, 52.05, 33.29, 24.11. HRMS (ESI) m/z: [M+H]^+^ Calcd for C_16_H_18_NO_4_^+^ 288.1231; Found 288.1229.

*Methyl 4-(5-fluoro-2-hydroxybenzoyl)-1H-pyrrole-2-carboxylate* (**3n**), a white solid (63%), R_f_ = 0.30 (n-Hexane/EtOAc, 8:2), m.p. = 126–127 °C. ^1^H NMR (400 MHz, DMSO-d6) δ 12.71 (s, 1H), 10.33 (s, 1H), 7.53 (s, 1H), 7.25 (t, J = 8.2 Hz, 2H), 7.09 (s, 1H), 6.96 (dd, J = 8.2, 4.3 Hz, 1H), 3.80 (s, 3H). ^13^C NMR (100 MHz, DMSO-d6) δ 194.54, 161.17, 158.83, 158.03, 134.93, 131.38, 130.24, 128.79, 124.57, 123.35, 120.89, 120.47, 56.83. HRMS (ESI) m/z: [M+H]^+^ Calcd for C_13_H_11_FNO_4_^+^ 264.0667; Found 264.0664.

*Methyl 4-(5-chloro-2-hydroxybenzoyl)-1H-pyrrole-2-carboxylate* (**3o**), a white solid (63%), R_f_ = 0.33 (n-Hexane/EtOAc, 8:2), m.p. = 123–124 °C. ^1^H NMR (400 MHz, DMSO-d6) δ 12.70 (s, 1H), 10.48 (s, 1H), 7.50 (dd, J = 3.3, 1.6 Hz, 1H), 7.41 (dd, J = 8.7, 2.7 Hz, 1H), 7.37 (d, J = 2.6 Hz, 1H), 7.08 − 7.04 (m, 1H), 6.98 (d, J = 8.7 Hz, 1H), 3.80 (s, 3H). ^13^C NMR (100 MHz, DMSO-d6) δ 194.14, 165.71, 160.16, 137.05, 134.92, 133.78, 132.83, 130.39, 128.79, 127.68, 123.80, 120.80, 56.83. HRMS (ESI) m/z: [M+H]^+^ Calcd for C_13_H_11_ClNO_4_^+^ 280.0372; Found 280.0376.

*Methyl 4-(5-bromo-2-hydroxybenzoyl)-1H-pyrrole-2-carboxylate* (**3p**), a white solid (60%), R_f_ = 0.32 (n-Hexane/EtOAc, 8:2), m.p. = 125–126 °C. ^1^H NMR (400 MHz, CDCl_3_) δ 11.84 (s, 1H), 9.74 (s, 1H), 7.91 (d, J = 2.4 Hz, 1H), 7.53 (dd, J = 3.1, 1.5 Hz, 1H), 7.49 (dd, J = 8.9, 2.4 Hz, 1H), 7.31 − 7.26 (m, 1H), 6.89 (d, J = 8.9 Hz, 1H), 3.86 (s, 3H). ^13^C NMR (100 MHz, CDCl_3_) δ 192.42, 161.52, 161.11, 138.25, 133.72, 127.78, 124.67, 124.31, 121.37, 120.43, 116.31, 110.43, 52.11. HRMS (ESI) m/z: [M+H]^+^ Calcd for C_13_H_11_BrNO_4_^+^ 323.9866; Found 323.9863.

The results of the X-ray diffraction analysis for compound **3g** were deposited with the Cambridge Crystallographic Data Centre (CCDC 2282072) (Appendix A).

## 4. Conclusions

In summary, we have developed a novel protocol for the synthesis of substituted pyrroles using DBU-driven sequences. This transformation involved the cleavage of one old C-C bond and the formation of two new C-C bonds in the process of aldehyde oxidation, Michael/retro-Michael decarboxylation, [3+2] cycloaddition, and aromatization. After investigating the generalizability of this cascade reaction, we found that the reaction performed well under optimized conditions, yielding a series of pyrroles in moderate to good yields. All synthesized compounds were tested using the MTT assay, and **3l** exhibited potent anti-colon cancer activity in vitro. Currently, we are making further efforts to develop efficient decarboxylation cycloaddition methods for the synthesis of privileged scaffolds.

## Data Availability

Data are contained within the article and Appendix A.

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
