# Peer review of "Metal-Free Catalyzed Oxidation/Decarboxylative [3+2] Cycloaddition Sequences of 3-Formylchromones to Access Pyrroles with Anti-Cancer Activity"

_molecules, 2023, doi:10.3390/molecules28227602_

Round 1
Reviewer 1 Report
Comments and Suggestions for Authors
Comments on the Quality of English LanguageThe quality of English is in general fine. Only two minor issues were detected:
· page 2, second paragraph under scheme 1: the sentence “In this study, we present a report on DBU-driven [3+2] cycloaddition sequences facilitate the synthesis of substituted pyrroles through a process of oxidation” should be rephrased
· page 5, last line, “and decarboxylation does not directly following the oxidation”, the sentence should be corrected
Reviewer 2 Report
Comments and Suggestions for Authors
[The reviewer is mainly concerned with the section of synthetic methods].
The present manuscript by Chongquing Univ. group describes an effective synthesis of 2,4-disubstituted pyrrole analogues through decarboxylative [3+2] cyclization of 3-formylchromones with commercially available TosMIC and methyl isocyanoacetate, useful and reactive bifunctional reagents. The regioselective synthetic method looks like useful and noteworthy to provide the substituted pyrroles as the phramacophere candidates. A comprehensive review of 3-formylchromones: Ghosh and Chakraborty ARKIVOC 2015, 288-361 and its ref. 29 (Tetrahedon 2008, 64, 5933) should be cited. In addition, carefully checked relevant reported reactions, especially for the use of commercially available 3-formylchromone as the Michael acceptor.
The author confirms the advantage of “metal free method” as depicted in the title, text, and Scheme 1. However, the reviewer considers that DBU (30 mol%) is considerably expensive and environmentally hazardous compared with highly accessible K2CO3 (30 mol%) (not transition metal base) (Table 1, entry 1). How is optimization of the use of K2CO3 using the MeCN solvent or another solvents? The author should offer a counterargument of the contradiction.
On the whole, the reviewer recommends the publication in Molecules including the comments and suggestions below after major revision.
<Comments and suggestions>
1. TOSMIC → TosMIC
2. Page 2, line 1; Decarboxylation → Decarboxylative cyclization
3. Scheme 1; “R” should be specified in Scheme 1; (R = p-toluenesulfonyl-, MeO2C-)
4. Page 2, lines 12-19; should be concisely illustrated in Scheme 1.
5. Page 3, line 1; DCE (dichloromethane) → DCE (1,2-dichloromethane)
6. Table 1; Blanc experiments without MW irradiation should be implemented to emphasis the method.
7. Text and Experimental section; The source of 3-formylchromone substrates 1a-1## and the carboxylic acid 4 should be addressed: known or unknown. In the case of unknown, the preparative procedure should be described in the experimental.
8. In the text, addresses “2-tosyloxy-4-benzoyloxy pyrroles” 3a-3k and “2-methoxycarbonyl-3-benzoyloxy pyrroles” 3l-3p for the readers’ easier comprehension.
9. Page 5, line 2; “the reported protocols for pyrrole synthesis” appropriate citation should be addressed.
10. Scheme 3; A shield tube apparatus was employed during the reaction (Experimental section). What is the oxidant for the conversion of 1a to 4 in the present reaction?
11. Page 6, line 1; For a possible mechanistic pathway, the English tense should be “present”: gave → give, formed → forms, etc.
12. Page 6, line 8; compound 10 → intermediate 10
13. Scheme 4, intermediate 10; the curry arrow should attack the N+
14. Experimental section; the weight of isocyanides and DBU should be inserted before the mmol, i.e. (## mmol) → (## mg, ## mmol).
15. Recommendation: X-ray structure of 3p is illustrated as a Figure.
Comments on the Quality of English Language
Some in appropriate presentations should be altered.
Reviewer 3 Report
Comments and Suggestions for Authors
In this paper, Chen and colleagues report a novel protocol for the synthesis of substituted pyrroles using a DBU-driven cascade reaction. The evaluation of the newly synthesized compounds as proliferation inhibitors against five different cancer cell lines was conducted. Here are some corrections and suggestions to consider:
1. In table 1, the time should be reported in minutes, as indicated in the information below.
2. Was the selection of bases based on their strength, or were other factors such as volume or the metal considered? Including this information would provide a better understanding of the role that the base plays in this reaction.
3. Since the core focus of the article is the use of DBU for driven decarbonylative cascades, it would be beneficial to delve deeper into the role played by this base in the reaction.
4. Could you provide an explanation as to why the assay with benzyl isocyanide did not yield the expected product?
5. In Scheme 4, it would be helpful to include all the missing mechanism reaction arrows for each step, as well as all the components involved in the reaction.
6. In the in-vitro description, it would be advantageous to include information about the control molecules. This would provide a better understanding of how well these compounds compare to those reported in the literature.
7. Ensure that the melting points of all the solids are included.
Comments on the Quality of English Language
Minor editing of English language required
Reviewer 4 Report
Comments and Suggestions for Authors
"Metal-free catalyzed oxidation/decarboxylative [3+2] cycloaddition sequences of 3-formylchromones to access pyrroles with anti-cancer activity," publication by Xue Li et al., provides an interesting new strategy in the synthesis of new pyrrole derivatives with a reasonably well performed mechanistic studies of the reaction. Moreover, biological activity was explored as well for the obtained compounds.
Generally, it is a well-written publication. I would recommend accepting this article for publication after the minor revision, if these points could be addressed/resolved:
Minor issues:
Page 1, in the text, an active compound, “pyoluteorin” is assigned for references 11 and 12, while reference 12 is for the “Pyrrolopyridinones”.
From page 3 and throughout the manuscript until the experimental section, degrees Celsius, "oC" something is wrong with formatting.
Scheme 2, regards 3g X-Ray data, it is very uncommon to add a picture of an X-Ray data to the scheme. Maybe it it sufficient that this information is in the supporting information. Of course you can always add this X-Ray of 3g as a separate high quality Figure in the manuscript.
Figure 2 seems to overlap with some of the text.
You should add the model and other specifications of HRMS spectrometer in the general part, and you must add all of the HRMS spectra to the supporting info.
You have provided 19F NMR spectrum of compound 3n, you should update general information part as well. Spectrometer, frequency, probe etc.
Experimental part compounds 3c, j, k, l, n, o "DMSO-d6", "d6" is more common in italics.
In the supporting info. "Figure S14. 1H NMR, 13C NMR and 19F NMR spectrum of 3n." plural is "spectra", please revise throughout.
One Major issue: compounds 3i and 3n contain a fluorine. In the experimental part of NMR spectroscopy there are some issues regards missing 13C NMR signals.
For instance in the compound 3i, you state that there are 17 signals, but in theory if there are no serious overlaps you should observe 18-20. In the supporting info, in the 13C i can observe that there are some signals in the range of 150-160 ppm not assigned.
In the compound 3n, you state that there are 14 signals but in theory if there are no serious overlaps you should observe 17-19.
You should add F,C couplings in Hz where possible.
Please revise.
But all in all.
This is a well written publication.
Best of Luck.
Comments on the Quality of English LanguageMinor revision is required.
Round 2
Reviewer 1 Report
Comments and Suggestions for Authors
The manuscript has been strongly improved, all suggestion were taken in consideration and performed. I now recommend the publication in Molecules after minor editing of English (see Comments on the quality of English)
Comments on the Quality of English Language- Page 3, line 85: "Microwave technology is a highly efficient heating technique for preparation bioactive compounds in organic chemistry", please coorect into "Microwave technology is a highly efficient heating technique for preparation of bioactive compounds in organic chemistry"
- Page 4, last line: "the model reaction were conducted with conventional heat at 100 ℃", please correct into ", the model reaction was conducted with conventional heating at 100 ℃"
-
Author Response
Dear Editor and Reviewer 1:
Thanks again! We appreciate very much for your valuable and helpful comments. The main corrections and responses are as following:
Comments and Suggestions for Authors
- Page 3, line 85: "Microwave technology is a highly efficient heating technique for preparation bioactive compounds in organic chemistry", please coorect into "Microwave technology is a highly efficient heating technique for preparation of bioactive compounds in organic chemistry"。
A: It was corrected.
- Page 4, last line: "the model reaction were conducted with conventional heat at 100 ℃", please correct into ", the model reaction was conducted with conventional heating at 100 ℃"
A: It was corrected.
Reviewer 2 Report
Comments and Suggestions for Authors
[The reviewer is mainly concerned with the section of synthetic methods].
The revised manuscript has been reviewed. My comments and suggestions are described bellow.
a) …… had been done by Ghosh and Chakraborty’s group [21-22]. → …… were reviewed by Ghosh and Chakraborty [21] and a relevant synthesis to the present our work was reported [22].
Along with the reviewer’s comment, the author should carefully cite the reported studies.
b) When the model reaction was conducted ……
The author does not provide reasonable responses to my argument. I consider that of the use of K2CO3 (30 mol%) compared with DBU (30 mol%) (not transition metal base) has significant advantage regarding total cost-effectiveness. Does the author have clear arguments? At least, the author should try the optimization effort the use of K2CO3. “Moreover, most of the starting materials 1a and 2b were left.” If it is the real result, the author should indicate the information in the text and/or Table 1.
> Response 4; I cannot detect the synthetic methods in the illustration in Scheme1: Only the duplication is shown.
> Response 10; the author should describe the comment in the text.
> Response 13; Zwitter ionic tautomer should be indicated. The – ion attacks the other – ion is eccentric in this scheme.
> Response 15; I am sorry for my mistake “X-ray structure of 3p.” The response of 4a is 3g?
Comments on the Quality of English Language
Author Response
Dear Editor and Reviewer 2:
Thanks again! We appreciate very much for your valuable and helpful comments. The main corrections and responses are as following:
The revised manuscript has been reviewed. My comments and suggestions are described bellow.
- a) …… had been done by Ghosh and Chakraborty’s group [21-22]. → …… were reviewed by Ghosh and Chakraborty [21] and a relevant synthesis to the present our work was reported [22].
Along with the reviewer’s comment, the author should carefully cite the reported studies.
A: Thank you for carefully checking, the original reported studies was cited in ref 21.
- b) When the model reaction was conducted ……
The author does not provide reasonable responses to my argument. I consider that of the use of K2CO3 (30 mol%) compared with DBU (30 mol%) (not transition metal base) has significant advantage regarding total cost-effectiveness. Does the author have clear arguments? At least, the author should try the optimization effort the use of K2CO3. “Moreover, most of the starting materials 1a and 2b were left.” If it is the real result, the author should indicate the information in the text and/or Table 1.
A: In terms of cost, K2CO3 has significant advantages DBU. However, when the model reaction was carried out with K2CO3 as a additive, quite low yield of desired product was observed and most of the starting materials were left (The results was described in Table 1. ). From reaction mechanism perspective, deprotonation of isocyanide is one of the key steps for affording products. To the best of our knowledge, PKa of DBU is 24.3 in MeCN. For K2CO3, PKa value is 9.3. So it indicated that deprotonation step is more favorable under the mixture of DBU and MeCN, just as the experiments shown in Table 1 and indicated in the manuscript with red.
> Response 4; I cannot detect the synthetic methods in the illustration in Scheme1: Only the duplication is shown.
A: Some typical examples for synthesis of pyrroles was indicated in the manuscript with red and corresponding references were cited in ref 18 and ref 19.
> Response 10; the author should describe the comment in the text.
A: It was corrected and describe in the text of page 6 with red.
> Response 13; Zwitter ionic tautomer should be indicated. The – ion attacks the other – ion is eccentric in this scheme.
A: It was corrected.
> Response 15; I am sorry for my mistake “X-ray structure of 3p.” The response of 4a is 3g?
A: We put the X-ray structure of 3g in Figure 2.